# Direct Measurement of Temperature Diffusivity of Nanocellulose-Doped Biodegradable Composite Films

**DOI:** 10.3390/mi11080738

**Published:** 2020-07-29

**Authors:** Hiroki Fujisawa, Meguya Ryu, Stefan Lundgaard, Denver P. Linklater, Elena P. Ivanova, Yoshiaki Nishijima, Saulius Juodkazis, Junko Morikawa

**Affiliations:** 1CREST—JST and School of Materials and Chemical Technology, Tokyo Institute of Technology, 2-12-1, Ookayama, Meguro-ku, Tokyo 152-8550, Japan; fujisawa.h.ac@m.titech.ac.jp; 2Reserarch Institute for Material and Chemical Measurement, National Metrology Institute of Japan (AIST), Tsukuba Central 3, 1-1-1 Umezono, Tsukuba 305-8563, Japan; 3Optical Sciences Centre and ARC Training Centre in Surface Engineering for Advanced Materials (SEAM), School of Science, Swinburne University of Technology, Hawthorn, VIC 3122, Australia; slundgaard@swin.edu.au; 4School of Science, RMIT University, Melbourne, VIC 3000, Australia; denver.linklater@rmit.edu.au (D.P.L.); elena.ivanova@rmit.edu.au (E.P.I.); 5Department of Physics, Electrical and Computer Engineering, Graduate School of Engineering, Yokohama National University, 79-5 Tokiwadai, Hodogaya-ku, Yokohama 240-8501, Japan; nishijima@ynu.ac.jp; 6Institute of Advanced Sciences, Yokohama National University, 79-5 Tokiwadai, Hodogaya-ku, Yokohama 240-8501, Japan; 7World Research Hub Initiative (WRHI), School of Materials and Chemical Technology, Tokyo Institute of Technology, 2-12-1, Ookayama, Meguro-ku, Tokyo 152-8550, Japan

**Keywords:** thermal properties, temperature diffusivity, nano cellulose

## Abstract

The thermal properties of novel nanomaterials play a significant role in determining the performance of the material in technological applications. Herein, direct measurement of the temperature diffusivity of nanocellulose-doped starch–polyurethane nanocomposite films was carried out by the micro-contact method. Polymer films containing up to 2 wt%. of nanocellulose were synthesised by a simple chemical process and are biodegradable. Films of a high optical transmittance T≈80% (for a 200 μm thick film), which were up to 44% crystalline, were characterised. Two different modalities of temperature diffusivity based on (1) a resistance change and (2) micro-thermocouple detected voltage modulation caused by the heat wave, were used for the polymer films with cross sections of ∼100 μm thickness. Twice different in-plane $\alpha_{\|}$ and out-of-plane $\alpha_{⊥}$ temperature diffusivities were directly determined with high fidelity: $\alpha_{\|}=2.12\times 10^{-7}$ m${}^{2}$/s and $\alpha_{⊥}=1.13\times 10^{-7}$ m${}^{2}$/s. This work provides an example of a direct contact measurement of thermal properties of nanocellulose composite biodegradable polymer films. The thermal diffusivity, which is usually high in strongly interconnected networks and crystals, was investigated for the first time in this polymer nanocomposite.

## 1. Introduction

Polymers are among the most widely used materials across the globe for the production and packaging of goods; there are very few industries that do not utilise polymers for their commercial outputs. Therefore, their production by simpler and more environmentally friendly methods is crucial. In this study, we used a newly developed starch–polyurethane polymer that is made by a simple low-temperature route [1]. We introduced nanocellulose (NC) at a concentration of 2 wt% as a dopant to the starch–polyurethane polymer. Cellulose is the most abundant material in nature that can be used for polymer fabrication. NC, or cellulose nanofibres (CNFs), were introduced as a building block for the production of more complex polymer composites with custom-designed electrical, optical, mechanical or electrical properties [2,3].

High-temperature diffusivity $\alpha =\kappa /(\rho c_{P})$, defined by the thermal conductivity κ [W/(m·K)], mass density ρ [kg/m${}^{3}$] and specific heat capacity $c_{P}$ [J/(kg·K)], is a necessary trait for optimised polymeric materials and their composites. As graphene has a very high in-plane 3000 W/(m·K) conductivity, it is a potential filler in polymeric and epoxy hosts for increased thermal and electrical conductivity; the out-of-plane conductivity of graphene-stack is only 5 W/(m·K) [4]. Composites that use graphene as the filler, it was previously shown that the thermal conductivity is increasing linearly with its weight percentage (up to 30 wt% before saturation), and in the case of NC-paper with 10 wt% of graphene, reached high 25 W/(m·K) values [5].

The thermal conductivity of these nanomaterials and their composites is usually measured by a non-contact flash method to determine temperature diffusivity α. During this measurement, one side of the sample is excited by an optical flash (such as a laser) onto an opaque absorbing (blackened) plane, while the temperature rise and its diffusion transport are detected by a pyrometer (non-contact) on the opposite side. The pyrometer detects the thermal radiation (irradiance) *J* of the black body emission according to the Stefan–Boltzmann (S-B) law $J=\epsilon \sigma T^{4}$, where ϵ is the emissivity of the object, σ is the S–B constant and *T* is the absolute temperature. Moreover, the specific heat capacity $c_{P}$ is usually measured by differential scanning calorimetry (DSC) for the known mass density of the sample according to $\kappa =\alpha \rho c_{P}$. However, for determination of nanoscale thermal properties (α,κ) of materials, there exists limitations to the flash method due to the need for a blackened absorptive transducer, well-known emissivity ϵ and a need of calibration.

Determination of temperature diffusivity α by a direct measurement is preferable for the determination of heat transport and focusing by nano-/microstructured materials exploiting long-range (hundreds of nm) ballistic phonon heat transport, which can be considerably more efficient on the nanoscale and can be directionally controlled [6]. Contact methods of measuring the thermal properties of materials experience a reduced sensitivity due to the large thermal capacitance of the micro-volume of the thermocouple brought into direct contact with the sample. In this study, we used miniaturised thermocouples and thermistors for direct measurements of α. Such measurements are in high demand to evaluate the variable thermal properties of nanomaterials and their composites, e.g., the typical thermal conductivity of protein-based polymers is 0.1 W/(m·K) [7], whereas α values exceeding that of metallic Cu (∼400 W/(m·K)) have been reported for drag-line silk (416 W/(m·K)) [8]. Exact determination of temperature diffusivity α is particularly important in micro-robotic applications [9] to control the volume-phase transitions, activated by light, harnessed for directional motion inside liquid. The link between optical, mechanical and thermal anisotropy in composite materials due to their microscopic (nanoscale) structure would benefit from direct measurement capabilities.

Here, we characterise the temperature diffusivity α of NC-doped starch–polyurethanes composite polymer films by the thermal wave method [10,11] with the heater and detector directly deposited onto the sample. Measurements of the in-plane and out-of-plane temperature diffusivity were made to reveal the anisotropy of thermal properties. This method is applicable for other polymer films and fibres with the sensor regions deposited using a simple shadow mask or defined by photo-lithography and lift-off.

## 2. Method: Thermal Wave

Temperature diffusivity α [m${}^{2}$/s] was measured by the thermal wave method [10], which determines the phase delay of a heat wave traversing the thickness *d* of the sample when the heat source is modulated at the frequency *f*:(1)$$\Delta \Theta =-\sqrt{\frac{\pi f}{\alpha }}d-\frac{\pi }{4}.$$

The amplitude of detected signal is defined by the thermal effusivity of the sample and substrate, *e* and $e_{s}$, respectively,
(2)$$Amp=A\frac{e}{\sqrt{f}{\left(e+e_{s}\right)}^{2}}e^{-kd},$$
where $k=\sqrt{(\pi f)/\alpha }$ and *A* is the empirical constant defined by experimental conditions. The temperature diffusivity α can be obtained from the Amp measurements; however, experimental detection of phase delay (Equation (Equation 1)) is simpler and potentially more reliable due to absence of additional error sources. This method was used to determine α values of popular polymers (on the macro-scale), to establish Δα changes upon glass transition and cold crystallisation [12], as well as in laser microstructured materials [13,14]. The phase-based thermal wave method as outlined by Equation (Equation 1) was used in this study.

The thickness of sample *d* should be larger than the diffusion length $L_{D}=\sqrt{\alpha /(\pi f)}$, i.e., a thermally thick sample condition. Temperature modulation ΔT(f) can be applied electrically using a resistor (see Figure 1) or optically by light absorbed on the surface of the sample and detected by either (1) electrical resistivity change of a resistor (the bolometer principle) or (2) by a thermocouple. Both modalities of the measurement were used.

## 3. Experimental

Nanocellulose (NC) fibres have a typical diameter of 2–5 nm and length between 44 and 108 nm when dispersed in water. NCs were incorporated into polymer films at 2 wt% concentration. The polymeric host for NC fibres was gelatinised starch and a polyhydroxyurethane mixture [15]. This green nanohybrid composite has a high 8.5 MPa tensile strength and reaches ∼30% elongation at breaking point due to the hydrogen bonding-enhanced network, a high melting temperature of ∼200 ${}^{\circ }$C and is 38% crystalline [15] (the procedure to determine crystallinity by X-ray diffraction is outlined in Section A.2). The nanocomposite polymer film with NC loading was made by a simple physical blending method. Briefly, starch, glycerol and water were added to a beaker and mixed to form a starch suspension with a final solid concentration of 5 wt% (w/w). After vigorously stirring at ∼95 ${}^{\circ }$C for 1 h, NC powder (at a loading of 2 wt%) was incorporated into the well-dispersed starch medium and stirring was continued for another 15 min. The mixture was sequentially homogenised with a T25 Ultra-Turrax (IKA-Labortechnik, Staufen, Germany) at 10k rpm for 2 min and sonicated with a Sonopuls ultrasonic homogeniser (Bandelin, Berlin, Germany) for 2 min to ensure an adequate dispersion of NC within the matrix. Afterwards, the homogenised dispersion was cast in plastic Petri dishes and the solvent was evaporated under ambient environment conditions.

Gold (Au) is heavily used in industry for the fabrication of electrodes, thermocouples and resistive heaters. The resistivity of Au is $\rho =2.44\times 10^{-8}\;\Omega \cdot$m, and a typical heater of length l=1 mm, width w=0.25 mm and thickness t=10 nm has resistance R=ρL/(wt)=9.8Ω (Figure 1). The actual Au films for heaters have resistance *R* of ∼50 Ω to match the output impedance of the common function generator and corresponded to ∼10-nm-thick coating.

The direct measurement of thermal diffusivity by the thermal wave method (Equation (Equation 1)) has become an industrial standard (ISO22007-3) using ai-Phase apparatus. It can be used for measurement of anisotropy in temperature diffusivity due to molecular alignment within the polymeric film. This method was recently used to demonstrate two orders of magnitude difference in α of polymeric spherulites of poly-l-lactic acid (PLLA) [16].

The calibration procedures for the measurement of thermal diffusivity α, thermal effusivity *e*, thermal conductivity κ and heat capacity per unit volume $c_{p}\rho$ have been established using calibration material with known heat capacity [11]; for the resistive heater and thermistor/sensor, a calibration with calibrated thermocouple is used. We used a heater and bolometer within their linear response range to the applied voltage (heater) and change of current/voltage (bolometer).

For the thermocouple sensor that detects temperature changes on the rear surface of the sample by a change in the electromotive force at the hot junction, the linearity of response has been established [13,17]. The sensor is wired via a cold junction with a copper lead, and it is assumed that the temperature change at the cold junction is negligible for the frequency of the measurement (a room temperature thermostat). Thus, the generated electromotive force is due to the temperature change at the hot junction and the detected electric signal is directly related to the temperature response from the sample. This type of sensor is capable of sensitive detection of minute temperature changes from micro-sized regions [17]. All the measurements were carried out within the linear response range.

## 4. Results

The experimentally measured phase delay of the temperature wave detected through the thickness *d* of the NC composite film is shown in Figure 2a. The heater is a 10 nm thick Au film with 0.5 mm width sputtered between Au contacts (40 nm thick) on one side of the sample, while the detector is placed on the opposite side.

For measurement of the temperature diffusivity through the sample (out-of-plane; Figure 1a), the detector was a 0.25 mm wide area of 10-nm-thick gold. For the in-plane (Figure 1b) measurement, an Au-Ni thermocouple was made on the polymer which was embedded into resin and sliced with a micro-tome (Buehler IsoMet, Manassas, VA, USA; top inset in Figure 2). Au and Ni thermocouples were made with 50×100μm${}^{2}$ stripes centred on the NC polymer. Examples of the evaporated Au-Ni thermocouples and Au termistors on a glass substrate are shown in Figure 2b with detailed scanning electron microscopy (SEM) close up images of the sensor regions. These sensors were deposited directly onto the polymer sample for the thermal wave measurements.

Direct measurement of the temperature diffusivity of the NC composite films was carried out with a very good signal-to-noise ratio and at a range of frequencies and linear fit (see Figure 2). The in-plane temperature diffusivity was ∼53% larger as compared with the out-of-plane diffusivity. A temperature diffusivity of $\alpha \approx 10^{-7}$ m${}^{2}$/s is typical for most polymers including biopolymers such as chitin, e.g., cicada wings have an in-plane diffusivity of $\alpha_{\|}=3.6\times 10^{-7}$ m${}^{2}$/s [18] and an out-of-plane diffusivity of $\alpha_{⊥}=0.7\times 10^{-7}$ m${}^{2}$/s. For comparison, the temperature diffusivity of air at normal conditions is $\alpha =2.17\times 10^{-5}$ m${}^{2}$/s, diamond $\left(7\pm 4\right)\times 10^{-4}$ m${}^{2}$/s and copper (Cu) $1\times 10^{-4}$ m${}^{2}$/s. The extremities in the temperature diffusivity of these different materials are separated only by three orders of magnitude, which is small compared with other properties such as electrical conductivity.

The NC composite films were inspected under cross-polarised light microscopy for the presence of optical/structural anisotropy. However, no birefringence was observed at visible spectral range. This is consistent with the high transparency of 0.2 mm thick polymer films (see Figure 3). The refractive index of a material is proportional to the mass density; therefore, birefringence is expected to also reflect the anisotropy in the mass density and could be linked to the packing density of a composite which has anisotropic constituents such as fibres. For example, the birefringence of silk fibre (one of the most birefringent biopolymers) is $\Delta n=1.7\times 10^{-2}$ [19] and that of NC $\Delta n=7.4\times 10^{-2}$ [20] at visible spectral range.

NC fibres were shown to increase the temperature diffusivity by approximately two times, even at very small (2 wt%) loading of NC. The same measurement technique can be used for other biopolymers and synthetic fibres (Section A.1) which are increasingly used in air/water filter applications and full characterisation of their thermal and mechanical properties are of paramount importance.

## 5. Discussion

The differences between the in-plane and out-of-plane temperature diffusivity are related to the structure of the polymer and molecular alignment of the building blocks. Polymeric fibres fabricated by electro-spinning and extrusion can exhibit increased thermal conductivity along the stretch direction. Thermal conductivity in a “stretched” polymer is higher as $\kappa \propto \sqrt{E}$, where *E* is the Young’s modulus. For example, in silk which has a ∼85% crystalline fraction [21], the beta-sheets are ordered directionally along the fibre, as directly measured by IR microscopy [22,23]. The largest temperature diffusivity for silk is expected along the fibre. Fibroin extracted from silk (depolymerised) and remade into amorphous silk film showed temperature diffusivity $1.6\times 10^{-7}$ m${}^{2}$/s [24]. Crystals aligned along the fibre direction underpin the mechanical and thermal properties of biopolymeric fibres and their optical properties can be determined with ∼20 nm resolution using scanning near-field optical microscopy (SNOM) [25] or atomic force microscopy (AFM) tip detection under thermal expansion of the surface illuminated at IR absorption bands [26]. It is noteworthy to add that the conductivity of air could influence the in-plane temperature diffusivity, especially on nano-micro rough surfaces such as cicada wings or when the interface contact is by mechanical attachment [18].

The temperature diffusivity of an epoxy resin with a greater (58 wt%) loading of NC was high at $5.9\times 10^{-7}$ m${}^{2}$/s [20] compared with our NC nanocomposite polymer sample at only 2 wt% of NC (see Figure 2). The anisotropy between the in-plane (high) and out-of-plane (low) thermal diffusivity of the epoxy nanocomposite film was approximately $\alpha_{\|}/\alpha_{⊥}=0.59/0.13\approx 4.5$ [20]. Here, the nanocomposite film had a mass density ρ=1.39 g/cm${}^{3}$, specific heat capacity at constant pressure $c_{p}=1.31$ J/(g·K) and thermal conductivity $\kappa_{\|}=1.1$ W/(m·K). As NC fibres have a very low coefficient of thermal expansion (CTE) of 0.1 ppm/K in the axial direction [20], composites with NC are promising for sealant applications at varying temperatures. NC increases the storage modulus of elasticity by 25% as compared with pure epoxy [20].

Another field of research where the direct measurement of thermal conductivity is important is the manufacture of thermoelectric materials, where the dimensionless parameter $ZT=S^{2}\sigma T/\kappa$ is used to determine the ability of charge flow while resisting the heat flow. Here, σ is the electrical conductivity and the thermopower or Seebeck coefficient S=−ΔV/ΔT. A temperature gradient ΔT applied to an electrically conductive material causes charge carriers to diffuse. The difference of charge concentration at the hot and cold ends defines a potential difference ΔV. The design of nanocomposites with high-ZT values are an active field of thermoplasmonics research where control of anisotropy is playing a large role [27]. Temperature diffusivity and thermal conductivity are key parameters which define the performance of perfect absorbers [28], which are defined by the ratio of optical energy absorbed during an optical cycle to that dissipated. The thermal properties of complex 3D polymeric photonic crystals could be directly measured using the proposed method [29,30,31]. Due to a high optical transmissivity of a thermistor made by 10 nm thick Au, it can be applied in direct temperature determination applications using laser tweezers where the micro-scale defines the challenges of direct measurement [32].

## 6. Conclusions

The temperature diffusivity of NC composite films for the in-plane $\alpha_{\|}=2.12\times 10^{-7}$ m${}^{2}$/s and out-of-plane $\alpha_{⊥}=1.13\times 10^{-7}$ m${}^{2}$/s were directly measured by the contact mode method. Measurements were carried out using samples with <200 μm cross sections. The same direct method using a thermocouple is applicable to polymer fibres.

## Figures and Tables

**Figure 1 micromachines-11-00738-f001:**
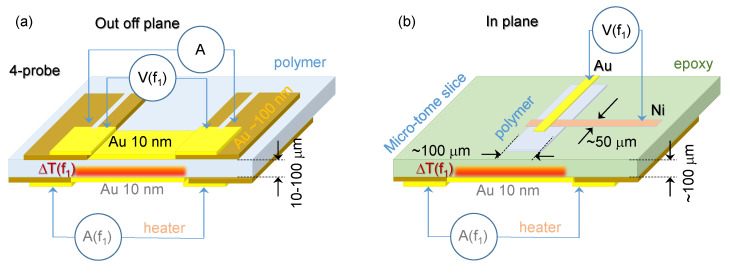
Two modalities of thermal wave measurements using detection by a resistance (**a**) and thermocouple (**b**) directly applied to the sample (polymer); the heater is sputtered directly on the opposite side of the sample. Temperature diffusivity in direction normal to the polymer sheet (out-of-plane) and along the plane (in-plane) can be measured. A micro-tome slice is used to prepare samples for the in plane heat transport measurements. The heat source is modulated at frequency $f_{1}$ = 4–36 Hz and detected by lock-in technique.

**Figure 2 micromachines-11-00738-f002:**
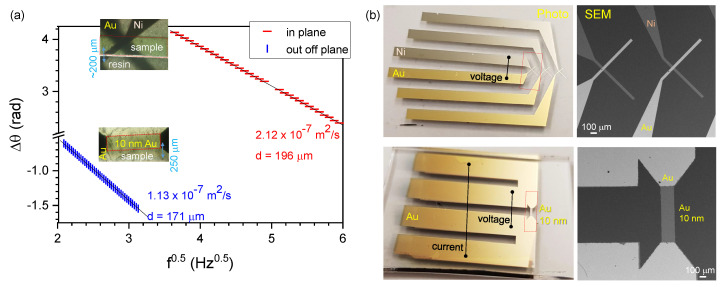
(**a**) Thermal wave Equation (Equation 1) measurements of the NC crystals (2 wt%) containing polymer across the sample (out-of-plane) and along the polymer film (in plane). Equation (Equation 1) is linearised by $\Delta \Theta \propto \sqrt{f}$ presentation. The thickness of the sample *d* was different. Insets are optical transmission micrographs of the measurement region. It is important to place the thermocouple within the projection cross section of the sample. (**b**) Photos show the thermocouple and resistance contacts evaporated onto a glass substrate (see Figure 1 for designation of the current and voltage connection ports). SEM images of the sensor regions.

**Figure 3 micromachines-11-00738-f003:**
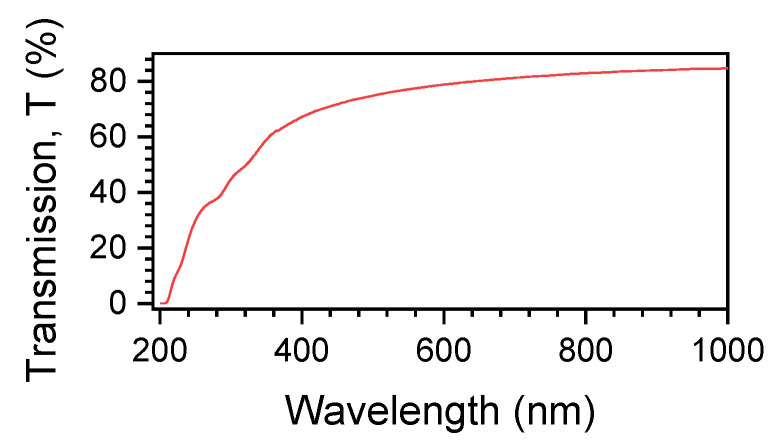
Optical transmission spectrum of 2 wt%. NC-containing polymer; thickness d=190±15μm.

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
