# Peer review of "Direct Measurement of Temperature Diffusivity of Nanocellulose-Doped Biodegradable Composite Films"

_micromachines, 2020, doi:10.3390/mi11080738_

Round 1

Reviewer 1 Report

The manuscript entitled " Direct measurement of temperature diffusivity of
nanocellulose composite films" will be beneficial for the nanocellulose research community. I have the following concerns that should be addressed in the revised manuscript.

  1. The title should include the name of the polymer matrix. In this work, the author used only 2wt% nitrocellulose. The remaining materials must be included in the title. 
  2. The introduction section should be rewritten as nothing was mentioned for the matrix (which is 98% of the polymer composite). 
  3. Experimental section: in this section reference 12 is under submission. therefore, the reviewer can not justify this section. Moreover, in line 76-77, melting point and crystallinity were mentions, but no information is available regarding these parameters. 
  4. Inline 90-91, anisotropy was referred for molecular alignment. Is it for any materials or for this polymer composite? If for this composite, then is it for nanocellulose or matrix or for both? Again, the detection of anisotropy for nanocellulose in the system is almost impossible as only 2wt% nanocellulose was used for this work. 
  5. line 134-141, it is expected that anisotropy regarding nanocellulose can not be detected (only 2wt% was used for composite film preparation). 
  6. Figure 4 (d), the WAXD of PET fiber did not confirm the crystallinity or amorphous percentage from the Debye-Scherrer diffraction ring. Both amorphous and crystalline materials can have similar types of Debye-Scherrer diffraction rings. The Debye-Scherrer diffraction ring only refers to the types of materials organization  (isotropic VS anisotropic). However, from the Debye-Scherrer diffraction rings, intensity VS 2theta plot can be extracted that may provide additional information regarding the crystallinity VS amorphous region. The authors must change this figure or provide additional information for extracting the crystallinity percentage from the Debye-Scherrer diffraction ring.

Author Response

Only critical remarks are answered. Changes are blue-color marked in the revision.

Remark. The title should include the name of the polymer matrix. In this work, the author used only 2wt% nitrocellulose. The remaining materials must be included in the title.

Answer. The composite was starch-polyurethanes-nanocellulose. We added “doped” in the title which convey the message of low concentration and still makes shorter and clearer title.

Remark. The introduction section should be rewritten as nothing was mentioned for the matrix (which is 98% of the polymer composite). More details about the matrix is added and appendix is added to define crystallinity determination.

Answer. Done. New reference ([1] in the revised version) is added with detailed description of the matrix.

Remark. Experimental section: in this section reference 12 is under submission. therefore, the reviewer can not justify this section.

Answer. Other reference is added where the same starch-polyurethanes matrix is described in details. Ref.: Assessment of interfacial interactions between starch and non-isocyanate polyurethanes in their hybrids, M Ghasemlou, F Daver, EP Ivanova, R Brkljaca, B Adhikari, Carbohydrate Polymers, 116656 2020 (it is ref [1] in the revised version).

Indeed nanocellulose is a small portion in the final material and the matrix characterisation is made in this newly added reference. We explicitly described protocol of fabrication, so all information is readily available. Appendix is added to serve a better sample description and crystallinity determination.

Remark. Moreover, in line 76-77, melting point and crystallinity were mentions, but no information is available regarding these parameters.

Answer. See the ref. above.

Remark.  Inline 90-91, anisotropy was referred for molecular alignment. Is it for any materials or for this polymer composite? If for this composite, then is it for nanocellulose or matrix or for both? Again, the detection of anisotropy for nanocellulose in the system is almost impossible as only 2wt% nanocellulose was used for this work.

Answer. This statement about anisotropy in molecular alignment is generic and is applicable for the matrix, dopand and their composite. The statement is validated with a reference at the end of this paragraph. It is important in discussion of the results and is introduced in the Experimental section. Also appendix is added to address crystallinity determination since this is one of the aspects of anisotropy and structural order/alignment.   

Remark. line 134-141, it is expected that anisotropy regarding nanocellulose can not be detected (only 2wt% was used for composite film preparation).

Answer. We discussed the experimental result of anisotropy measured experimentally and we do not assign it to nanocellulose as a definite fact. The key point of this study is that a direct measurement of anisotropy of thermal properties is possible for sub-100 micron materials.  

Remark.  Figure 4 (d), the WAXD of PET fiber did not confirm the crystallinity or amorphous percentage from the Debye-Scherrer diffraction ring. Both amorphous and crystalline materials can have similar types of Debye-Scherrer diffraction rings. The Debye-Scherrer diffraction ring only refers to the types of materials organization  (isotropic VS anisotropic). However, from the Debye-Scherrer diffraction rings, intensity VS 2theta plot can be extracted that may provide additional information regarding the crystallinity VS amorphous region. The authors must change this figure or provide additional information for extracting the crystallinity percentage from the Debye-Scherrer diffraction ring.

Answer. Thank you for the good point. We made a deeper study and added description of the method in appendix. From the data we estimated 30% volume fraction of crystalline phase in the PET fiber. This deeper analysis hinted us to make a dedicated study of PET which we can have in very different streached films crystalline and amorphous as well as in the fiber form. PET data were used in this manuscript for comparison and makes the entire study less focussed. As required by Referee 2, we rearrange material to have one focus on new polymer doped with nanocellulose. The crystallinity determination which is very important factor is moved to the appendix. 

Reviewer 2 Report

Fujisawa et al. reported the manuscript entitled ‘‘Direct measurement of temperature diffusivity of nanocellulose composite films.’’ This manuscript needs major revision before publication. Some comments are as follow:

  1. The introduction section should be a comparative study with some nanocellulose-based composites articles. Please mention the novelty of the current study.
  2. In line 80 authors need to check carefully. Change from 95oC to 95 oC maintain consistency. In line 4, 2% wt or 2wt% check it.
  3. If possible the authors need to provide thermos gravimetric analysis (TGA) data of the fabricated samples.
  4. Please rearrange Fig. 3 in a better way.
  5. The author should give scanning electron microscopy (SEM) images, it is suggested to use an arrow mark for the location of nanocellulose in the matrix.
  6. Regarding Fig. (4b, 4c, and 4d) explanations are not clear need more explanations.

Author Response

1. The introduction section should be a comparative study with some nanocellulose-based composites articles. Please mention the novelty of the current study.

Answer. Novelty is in characterisation of the newly developed material. We also take out result on fiber polymer to better focus the results of this study. Possibility to make measurements on fibers is transferred to appendix.  Text is rephrased and re-edited. 

2. In line 80 authors need to check carefully. Change from 95oC to 95 oC maintain consistency. In line 4, 2% wt or 2wt% check it.

Answer. Thank you. Corrected. wt% is better version as it is in spoken language.

3. If possible the authors need to provide thermos gravimetric analysis (TGA) data of the fabricated samples.

Answer. It is very valid point. We currently cannot access labs in Melbourne and Tokyo, hence, we cannot add those results. Discussion is extended based on some earlier results.

4. Please rearrange Fig. 3 in a better way.

Answer. Figure 3 is updaded to make it more detailed and clear.

5. The author should give scanning electron microscopy (SEM) images, it is suggested to use an arrow mark for the location of nanocellulose in the matrix.

Answer. We have added SEM images of the termistor and thermocouple. These are locations where polymer is measured since those structures are deposited directly onto the sample. Unfortunately we cannot measure SEM of the polymer sample now. From earlier results we know that nanocellulose is not distinguishable in a polymer matrix, especially, at low loading.  

6. Regarding Fig. (4b, 4c, and 4d) explanations are not clear need more explanations.

Answer. Former Figure 4 is simplified and moved to appendix since data on PET can be seen as not related to the new polymer investigated in this study.